# Survey of Pain Knowledge and Analgesia in Dogs and Cats by Colombian Veterinarians

**DOI:** 10.3390/vetsci6010006

**Published:** 2019-01-10

**Authors:** Carlos Morales-Vallecilla, Nicolas Ramírez, David Villar, Maria Camila Díaz, Sandra Bustamante, Duncan Ferguson

**Affiliations:** 1Facultad de Ciencias Agrarias Universidad de Antioquia, Medellín 050010, Colombia; cmoralesvallecilla@gmail.com (C.M.-V.); nicolas.ramirez@udea.edu.co (N.R.); mcamila.diaz@udea.edu.co (M.C.D.); smilena.bustamante@udea.edu.co (S.B.); 2Department of Comparative Biosciences, College of Veterinary Medicine, University of Illinois at Urbana-Champaign, Urbana, IL 61802, USA; duncan.c.ferguson@gmail.com

**Keywords:** analgesia, dogs, cats, epidemiological survey, Colombia

## Abstract

A questionnaire study was conducted among 131 veterinarians practicing in the city of Medellin, Colombia, to assess views on pain evaluation and management in dogs and cats. When pain recognition and quantification abilities were used as a perceived competence of proper pain assessment, only 83/131 (63.4%, confidence interval (CI) 0.55–0.72) were deemed to have satisfactory skills, with the rest considered to be deficient. There were 49/131 (37.4) veterinarians who had participated in continuing education programs and were more confident assessing pain, with an odds ratio (±standard error) of 2.84 ± 1.15 (*p* = 0.01; CI 1.27–6.32). In addition, the odds of using pain scales was 4.28 ± 2.17 (*p* < 0.01, CI 1.58–11.55) greater if they had also participated in continuing education programs. The term multimodal analgesia was familiar to 77 (58.7%) veterinarians who also claimed to use more than one approach to pain control. Nevertheless, homeopathy was the preferred alternative approach in 71/77 (92%). There were major misconceptions on side effects and/or contraindications for use of opioids and non-steroidal anti-inflammatory drugs (NSAIDs) by most veterinarians. In addition, the lack of multimodal analgesia by at least 40% of the practitioners, combined with heavy reliance on weak analgesics (i.e., tramadol) or those with no proven record of efficacy (homeopathic remedies), denotes major deficits in education at the undergraduate level and a need for additional continuing education designed to fulfill the gaps in knowledge identified in this study, and overcome ideological convictions not supported by scientific evidence.

## 1. Introduction

The effective management of pain relies on the ability of veterinarians to recognize and measure it in a reliable manner. Pain continues to be largely measured based on subjective measurements of behavioral signs. However, numerous advances have been made in the last two decades in the ability to accurately diagnose and treat painful conditions, based on a better understanding of how pain develops and is perpetuated. These have been put together in guidelines such as those of the World Small Animal Veterinary Association (WSAVA) for the recognition, assessment and treatment of pain [1]. Similar and more contemporary documents developed by a collaborative effort of the American Animal Hospital Association (AAHA) and the American Association of Feline Practitioners (AAFP) have been published [2], incorporating into earlier versions of the same document improvements reflecting continuing developments in research, technology, and expert opinions [3].

Despite all advances and attempts to develop global guidelines, standardization efforts are often limited by differences in attitude, education, and analgesic modalities available in different geographical regions and nations. In many developing countries, including Colombia, common analgesic drugs may be unavailable or hard to acquire by veterinarians. When availability is not an issue, expense and/or regulatory legislation guidelines may limit use of certain, particularly non-generic ones. In addition, education at the undergraduate level in Colombia is usually provided by professors that have not had many opportunities to receive education abroad in more advanced veterinary institutions.

Epidemiological surveys have been helpful at identifying factors limiting the development and application of new analgesic techniques and treatments. Such studies have been published in different countries from the early 1990s, including in the USA [4], Canada [5,6], UK [7], France [8], Finland [9], and Brazil [10]. Through repeat surveys, some of these studies revealed important positive changes and trends in pain management over time. For example, the Canadian studies showed that the percentage of dogs and cats receiving analgesia after castration and ovariohysterectomy was increased about 3-fold from 1994 to 2001 [5,6]. A more recent survey conducted among UK veterinary surgeons also showed that analgesic prescriptions had increased from previous surveys [11]. A variety of reasons were attributed to these changes, including greater awareness of pain, greater knowledge of how to treat pain, more opportunities for continuing education, and greater availability of non-steroidal anti-inflammatory drugs (NSAIDs).

At present, there is no information on the attitude and knowledge of veterinarians working in Colombia towards providing adequate analgesia in small animal practice. Thus, the objectives of this study were to assess the attitudes of veterinarians on the topic of pain relief and to identify gaps in knowledge of analgesic usage that should be addressed to increase the well-being of animals undergoing surgical procedures. The information gathered serves to guide future research and continuing education in this area.

## 2. Materials and Methods

### 2.1. Questionnaire

A questionnaire was developed based on the one designed previously by the French Group of Clinical Research on Veterinary Analgesia (GRCAV) and kindly provided by Dr. Marine Hugonnard [8]. In order to increase responses and reduce bias due to question ambiguities or difficulties in completing the questionnaires, veterinarians practicing in the city of Medellin were personally interviewed by two of the veterinary investigators during 2017. Ethics approval was granted by the University of Antioquia Ethics Review Board and the participants were assured of the confidentiality of the results. We propose that compared to mailed questionnaires, personal interviews should reduce personal bias of the respondents. The sample size of 131 veterinarians was randomly chosen from a finite population of about 400 listed small animal practitioners working in Medellin. The dates for the interviews were scheduled according to the veterinarians’ personal agenda with 100% response rate. Their names and institution would remain anonymous at the completion of the study and if it became publishable. The practitioners were surveyed to assess their views on pain evaluation and control in dogs and cats. Briefly, the questionnaire consisted of 5 sections to collect information as follows: (1) demographics; (2) pain assessment (level of concern, ability to recognize and quantify pain, and type of pain indicators used); (3) competence to control pain and reasons for the use of analgesics; (4) extent of use of different analgesic classes (opioids, non-steroidal anti-inflammatory drugs, dissociative agents, α2-agonists); and (5) depth and quality of veterinary teaching on the topic of pain and interest on continuing education on the subject.

As the ability to manage pain adequately cannot be assessed properly through this type of study, for the purpose of statistical analysis, the term “pain assessment” was arbitrarily deemed adequate when both “recognition” and “quantification” were judged to be excellent or good, and inadequate if one of these abilities was deficient, poor or mediocre. In order to rank what veterinarians considered to be the most useful pain indicators, they were asked to choose three options from a list of nine possible descriptors, for both dogs and cats separately. However, it must be emphasized that in a clinical setting, most pain score scales use a composite of the descriptor options provided here, and these are usually ranked numerically according to their associated pain severity. Consequently, a question on whether pain scales were used to assess acute pain was also included in the questionnaire. Participants were asked to rank a list of 5 general reasons for initiating pain treatment in order of decreasing importance, with 1 considered the most important and 5 the least important.

The extent of use of different classes of analgesics (including adjuvants) was also recorded separately for dogs and cats, by inquiring about the frequency at which individual drugs from a list of 52 were used. Although corticosteroids are not considered as primary analgesic drugs, they were also included as they can exert a pain-modifying effect by reducing inflammation. To narrow the type of medications used for surgical procedures, they were asked to name the most common type of analgesics used in the pre- and postoperative stages. This was followed by asking whether they have heard and read about the use of “multimodal analgesia”. When opioid use was acknowledged, the participant was then asked to select two out of 12 likely side effects, with some misleading/confounding descriptors, considering dogs and cats separately. The responses were used to evaluate knowledge of potential side effects. Similarly, knowledge of contra-indications for using NSAIDs was gauged by asking the participant to designate which of 10 proposed conditions were significant contraindications. To conclude the questions on the section for pain control, the participant was asked to denote their use of alternative modes of analgesia including homeopathy, acupuncture, pulse electromagnetic therapy, neural therapy, physical therapy and/or herbal remedies.

Finally, the quality of prior veterinary school instruction on the topic of pain and the individual’s potential interest in continuing education were addressed. Three questions addressed the depth and quality of veterinary undergraduate education on pain recognition and management as well as their subsequent participation in continuing education courses on the topic.

### 2.2. Statistical Analysis

All data collected through questionnaires were entered into Excel worksheets (Microsoft Corp. Redmond, WA, USA) and then exported to Stata 12.0 (StataCorp, 2011, College Station, TX, USA) for analysis. The data were examined for erroneous entries and if any were found, they were removed or corrected. Descriptive statistics were computed for all variables of interest. A chi-square or exact Fisher test were performed using a 5% level of significance, as an association test for the variables of interest.

## 3. Results

Demographic characteristics are shown in Table 1. The majority of veterinarians worked in clinics that, as opposed to hospitals, do not provide around-the-clock intensive care. Most veterinarians (101 out of 131) had graduated from the University of Antioquia located in Medellin. There were more males (58%) than females (42%) and the dominant age category was young graduates with less than 10 years in practice.

### 3.1. Pain Assessment

A diagram showing the responses for pain assessment is shown in Figure 1. The level of concern over pain recognition and management was high in the majority of the interviewed veterinarians, with only 12 (9.3%) showing moderate interest. Most veterinarians felt their skills in recognizing pain were better than quantifying the degree of pain. When considering only good and excellent skills for either ability as a perceived competence of proper pain assessment, only 83/131 (63.4%, CI 0.55–0.72) were deemed to have satisfactory skills on pain evaluation, with the rest (49/131, 36.6%) considered to be deficient (Figure 1). No association was found between gender and pain assessment (χ^2^ = 0.096, *p* = 0.7). Similarly, there was no association between years in practice (χ^2^ = 1.23, *p* = 0.5) or postgraduate degrees attained (χ^2^ = 0.0, *p* = 0.99) with pain assessment. However, 49/131 (37.4%) veterinarians had participated in some type of continuing education program (i.e., seminars, conferences, webinars), and were more confident assessing pain, with an odds ratio of 2.84 (*p* = 0.01, 95% CI 1.27–6.32) for adequate pain evaluation. In addition, the odds of veterinarians using pain scales was 4.28 ± 2.17 (*p* = 0.004, CI 1.58–11.55) greater if they had also participated in continuing education programs.

The rank order of pain indicators considered most useful in dogs and cats by the participants is shown in Table 2. The preferred option for dogs was the adoption of abnormal postures with 89 (68%) times cited, whereas changes in behavior that included lack of grooming were the most cited selection for cats with 82 (64%) responses. These were followed by changes in mobility and response to manipulation in dogs, and by abnormal postures and changes in mobility in cats. For the variables that differed most between dogs and cats, response to manipulation was an accurate determinant of pain assessment in dogs (χ^2^ = 4.31, *p* = 0.04) but not in cats; however, inappetence was selected three times more in cats than dogs.

Of the five reasons suggested for treating pain, the ones selected in decreasing order of importance were: (1) to alleviate pain and suffering; (2) to mitigate adverse effects of pain on body organ/systems; (3) to reduce postoperative complications and time of recovery; (4) to allow for therapeutic manipulations; and (5) to satisfy owner concerns.

### 3.2. Pain Control

On average, the respondents considered their use of analgesics to be better in dogs than cats. The majority of veterinarians estimated their analgesic use was good (96/131, 75%) or excellent (13/131, 10%) in dogs. For cats, the majority considered their use was good (86/131, 66%) and only a few felt it to be excellent (8/131, 6.2%). In cats, 34 (27%) believed their use was poor or mediocre, as compared to only 19 (15%) in dogs.

As depicted in Figure 2 and in order of decreasing frequency of use, the most commonly prescribed NSAIDs in dogs were: meloxicam (93.1%), dipyrone (85.5%), carprofen (78.6%), ketoprofen (75.6%), flunixin (53.4%), etodolac (37.4%), and ketorolac (13.7%). Interestingly, non-recommended NSAIDS like naproxen and ibuprofen were cited 14 (10.6%) and 4 (3%) times, respectively. However, the more recently marketed “coxib” category of NSAIDs, like mavacoxib and firocoxib, were only used by 11 (8.4%) and 3 (2.3%) veterinarians, respectively. In cats, all NSAIDs were cited less frequently than dogs, and only 4 NSAIDs were primarily used as follows: meloxicam (89%), dipyrone (63.3%), ketoprofen (53.4%), and carprofen (24.4%). There were 10 (7.6%) veterinarians who also selected etodolac in the list of prescribed NSAIDs for cats (Figure 2).

To assess knowledge on common contraindications for use, and/or adverse effects produced by NSAIDS in dogs and cats, a list of 10 proposed conditions were provided, and they were asked to consider the use of an NSAID as contraindicated or not. In order of decreasing frequency, the conditions selected as contraindications were: gastric ulcers (97%), renal insufficiency (85%), concomitant use with glucocorticoids (62%), concomitant use with other NSAIDs (47%), anticoagulant effects (35%), and thrombocytopenia (18%). Conditions not proven to be attributed or caused by NSAIDs were answered correctly by 95% of the veterinarians, including: asthma, carcinogenicity, hypothyroidism, and leukemia.

For opioids, as for NSAIDs, the percentage of users in dogs was higher than in cats for most drugs cited, but the frequency order of drugs chosen was similar in both species (Figure 2). Tramadol was the most popular drug prescribed by ≥80% veterinarians, and this was followed by morphine, fentanyl, hydromorphone, and butorphanol. The least used opioids were buprenorphine, meperidine, methadone and oxymorphone.

When asked to select 2 options among 12 possible side effects of opioids, the greatest concerns were respiratory depression and hypotension (Table 3). Typical side effects produced by the most commonly used pure mu opioid agonists in surgical procedures such as morphine and hydromorphone (i.e., panting > vomiting > ileus > dysphoria; bradycardia for fentanyl) were not recognized, and instead, many confounder descriptors were preferred by most veterinarians, indicating lack of proper knowledge on this topic and erroneous extrapolation from human medicine. For example, sedation that is a desired side effect for surgical procedures and that typically accompanies “analgesia” of pure opioid agonists, was not recognized as a typical side effect and was only cited 15 (12.2%) and 12 (9.2%) times for dogs and cats, respectively.

The frequency of use of other analgesic drugs questioned, that included glucocorticoid anti-inflammatory drugs (although these are not typically considered as primary analgesics), were: dexamethasone (84% dogs, 75.5% cats), prednisolone (79.4% dogs, 64.9% cats), methylprednisolone (57.3% dogs, 35.9% cats), betamethasone (51.9% dogs, 31.3% cats), triamcinolone (36.6% dogs, 27.5% cats), detomidine (94.6% dogs, 47.3% cats), xylazine (64.1% dogs, 47.3% cats), dexmedetomidine (6.1% dogs, 4.6% cats), ketamine (72.5% dogs, 73% cats), lidocaine (89.3% dogs, 70.2% cats), bupivacaine (38.9% dogs, 21.4% cats), benzocaine (17.5% dogs, 5.3% cats), and procaine (81.6% dogs, cats 89.3%).

The term “multimodal analgesia” was known by 77 (58.7%) of the 131 veterinarians surveyed, affirming their likelihood of using multiple simultaneous approaches to control pain. The remaining veterinarians typically used one medication (NSAIDS or opioid) or an alternative approach to control pain. The different alternatives used for pain management in decreasing order of frequency were: homeopathy (71/77, 92%), pulse electromagnetic therapy (42/77, 55%), acupuncture (25/77, 32%), neural therapy by local anesthetic injections (35/77, 45%), chiropractics (22/77, 28%), and use of herbal remedies (22/77, 28%). Except for homeopathy and herbal use, these modalities were generally provided by referral to specialty clinics.

In order to narrow the type of analgesics used in the pre- and post-operative periods, participants were asked to name the most common drugs used in specific surgical procedures for dogs and cats. The 6 most common drugs used during the pre-operative period in dogs were: tramadol (58%), meloxicam (47.3%), acepromazine (35.8%), ketamine (17.5%), xylazine (17.5%) and dipyrone (16%). For the post-operative period, the most common drugs used were: meloxicam (80%), tramadol (62.5%), ketoprofen (29%), and dipyrone (25%). In cats, the most frequent drugs mentioned for the pre-operative period were: tramadol (50%), meloxicam (40%), acepromazine (24%), ketamine (21%), and ketoprofen (14.5%). For the post-operative period and in decreasing order of frequency they were: meloxicam (73%), tramadol (53%), ketoprofen (26%), dipyrone (15%), and morphine (13%).

Finally, the majority of veterinarians declared their undergraduate education pertaining to recognition of pain was inadequate or deficient (100/131, 76.3%), and a similar number affirmed they were not properly taught to treat painful conditions.

## 4. Discussion

During the personal visits and interviews of participating veterinarians, we discerned high interest in the topic of pain management in part through the high participation and collaboration rate. Given that this was a personal interview, it is possible this might have biased the responses against saying that there was low interest. In our survey population, approximately 3 out of 4 veterinarians working in the city of Medellin were graduates of the University of Antioquia. As other Colombian universities were not equally represented here, part of the information gathered, such as deficiencies in the undergraduate curriculum were likely skewed to more highly represent graduates of this institution, and may not be representative of the quality of education at other veterinary schools. However, we felt that this potential educational bias was unlikely to alter most other responses of the questionnaire.

In general, veterinarians felt more confident in pain recognition (89.3%) than quantification (63.3%), both crucial factors for choosing appropriate analgesics and assessing their effects. The most important determinant of whether they considered their skills for pain assessment to be adequate was their prior participation in continuing education programs; in addition, such participation was associated with greater use of pain score scales. Veterinarians expressed more confidence using analgesics in dogs (85%) than cats (72.2%), which most likely explained the higher frequency of use for most analgesics in the former species. However, the frequency of analgesic use should also be tempered by determination of appropriated usage. Two Canadian surveys that had included specific information on different categories of surgery (i.e., orthopedic versus ovariohysterectomy) and therapeutic regimens (i.e., type of analgesic, dosage, route of administration, number of doses and interval) concluded that, even when most animals received analgesics in the perioperative period, neither the drugs chosen nor the duration of treatment were optimal [5,6]. In particular, butorphanol, rather than pure mu opioid agonists or NSAIDs, was the most commonly used drug for all type of surgeries examined. This raises concern because it is considered a mild analgesic with a short duration of action (≤ 2 h) in dogs and cats. Similarly, another concern was the general use of only one analgesic drug to control pain. In our survey, only 58.7% (77/131) veterinarians were familiar with the term “multimodal analgesia” and, even when they used more than one approach to pain control, the preferred additional method was the use of homeopathic remedies (71/77, 92%). To our knowledge, the efficacy of homeopathic remedies is largely unproven, and in spite of being quite controversial, there are no standard clinical and placebo-controlled trials to substantiate any claims of benefit. This finding is regrettable, as the Colombian veterinarian typically relies on commercial literature, personal judgment from anecdotal evidence, and advice from senior colleagues to base their decision-making process in clinical practice. In addition, over three-quarters of the veterinarians surveyed did not consider their undergraduate training to be adequate regarding the recognition and treatment of pain in animals. We would argue that the role of veterinary education is to instruct students to take an evidence-based approach and develop clinical critical thinking in their decision-making processes as future professionals. In the latest guidelines for the management of pain in dogs and cats, the American Animal Hospital Association (AAHA) and American Association of Feline Practitioners (AAFP) discourages the use of homeopathy for the treatment of pain, literally stating “sole reliance on homeopathy to treat painful conditions is, in essence, withholding pain treatment” [2]. Again, this emphasizes the importance of teaching Colombian veterinary students and professionals in practice to use reliable sources of information in their decision-making process.

With regards to which pain indicators were considered more useful to recognize pain, our survey participants ranked abnormal postures, mobility, and response to manipulation as the most important signs in dogs. By contrast, the absence of normal behavior or taking on unusual behaviors was considered the most valuable indicator of pain in cats. In addition, inappetence was chosen 3 times more in cats than dogs. Although the rank of the most valuable pain indicators for pain recognition was similar to those reported in a French and Australian survey [8,12], it should be highlighted that in clinical practice most acute pain score scales use a composite of the descriptor options provided here, and these are usually ranked numerically according to their associated pain severity. For example, the Glasgow Composite Pain Scale includes “vocalization” as a measure of acute pain to be scored in intensive care patients. In our questionnaire, kindly provided by the authors of a French survey [8], the preferred pain indicators were similar to those of French veterinarians, with options like “vocalization” rarely chosen. Poor reliance on vocalization as a pain indicator compared to other ones may be that vocalization can also be strongly associated with animal distress; so, in order to answer questions like this one through a questionnaire, maybe the clinical setting in which the patient is evaluated should also be described. For example, the use of any opioid can result in dysphoria, a syndrome where the animal shows vocalization and extreme agitation that can be difficult to distinguish from pain. In this situation, dysphoria would be caused by the very treatment we are giving in attempts to provide pain relief.

The most commonly used opioids and NSAIDs, and their frequency of use, was almost identical to that reported by a recent Brazilian survey [10]. Tramadol and meloxicam were the most widely used drugs within each category in both surveys. This differs from some North American and European surveys, where butorphanol and/or buprenorphine appear to be the most popular opioids [5,6,7]. In Colombia, as it may happen in Brazil, most strong opioids are hard to acquire or cost-prohibitive. Consequently, tramadol was the most prescribed opioid as it can be attained over-the-counter and is not a controlled drug. However, tramadol is a weak analgesic and for moderate to severe pain should ideally be part of a multimodal analgesia therapy. For example, when the antinociceptive effects of tramadol were evaluated in cats using mechanical and pressure models, there was little or no effect on nociceptive thresholds at 1 mg kg^−1^ subcutaneously, suggesting it may have limited analgesic effect in cats [13]. However, when combined with acepromazine, the combination did enhance analgesia above that of either drug alone [13]. A recent study using pressure platform gait analysis showed that tramadol (5 mg kg^−1^ orally every 8 h) provides no clinical benefit for dogs with osteoarthritis of the elbow or stifle joint [14]. Another study showed that when tramadol was used in the preoperative period of ovariohysterectomy, and at higher doses of 4 mg kg^−1^ subcutaneously, it improved the postoperative comfort levels of cats since rescue analgesia was not necessary compared to pethidine [15]. When combined with the NSAID vedaprofen 1 h before ovariohysterectomy, tramadol (2 mg kg^−1^ subcutaneosuly) provided more effective postoperative analgesia and prevented hyperalgesia than when either drug was used on their own [16]. In general, all these studies suggest that tramadol is likely to provide added analgesia when combined with other drugs but should not be used alone for controlling pain. In addition, a decrease of 60 to 70% in tramadol plasma concentrations occur after just 8 days of repeated administrations in dogs, making its long-term efficacy at best questionable [17].

When asked about opioid and NSAIDs side effects, a few misconceptions were identified. For opioids, respiratory depression was the main side effect that concerned veterinarians, an extrapolation from human medicine as dogs and cats are not sensitive to this effect at therapeutic dosages unless the animal is markedly mentally depressed [18]. The second side effect that concern most veterinarians was hypotension. Except for dissociative agents, hypotension is the main adverse side effect of general anesthetics, and since opioids are commonly used in the perioperative period of surgical procedures, it is possible they attributed this effect to opioids. When morphine is administered intravenously as a bolus, particularly in cats, it can lead to histamine release and hypotension [19]. However, this is not typically reported for other opioids and most have minimal effects on cardiac output and mean arterial pressure at clinically relevant doses.

With regards to NSAIDs, commonly used human drugs such as ibuprofen and naproxen have a narrow safety margin in dogs and cats and should not be used when other approved and easily accessible NSAIDs can be acquired [20]. Etodolac is only prescribed for use in dogs and yet, 10 (7.6%) veterinarians indicated they prescribed it for cats. Although, most veterinarians responded correctly that the main side effects of NSAIDs were gastric ulcers and potential for nephrotoxicity [21], it was most troublesome to find that only 47% of the veterinarians considered as a contraindication the co-administration of an NSAID with a corticosteroid. This demonstrated a poor knowledge of contraindications for opioids and NSAIDs. In the French survey that used the same questions on adequate knowledge of NSAIDs, the correct answer by the expert panel was that concurrent use of a NSAIDs with glucocorticoids was absolutely contraindicated, as opposed to relative or not contraindicated [8]. In reviews on general guidelines for use of NSAIDs, this group of analgesics should not be administered to patients with acute kidney injury (AKI), hepatic insufficiency, conditions associated with low “effective circulating volume”, coagulopathies and evidence of gastrointestinal disorders of any kind [18]. In addition, two or more NSAIDs should not be administered concurrently, as they may act synergistically to cause gastric lesions.

## 5. Conclusions

Taken all together, the above findings suggest that veterinarians may not be providing adequate analgesia as major gaps exist in the knowledge of analgesia and pain assessment. The extent to which pain therapy is also provided by other means (i.e., physical therapy, controlled exercise, acupuncture) should also be explored in future surveys, as analgesia should not just revolve around drugs. Because significant concern and interest for adequate pain management was expressed by veterinarians, the potential for future educational progress is encouraging. It also provides a constructive environment for continued improvement in the area of pain evaluation and management. Finally, as veterinarians were all registered within the urban area of Medellin, it will be of interest to extend this study to other cities and rural areas of Colombia.

## Figures and Tables

**Figure 1 vetsci-06-00006-f001:**
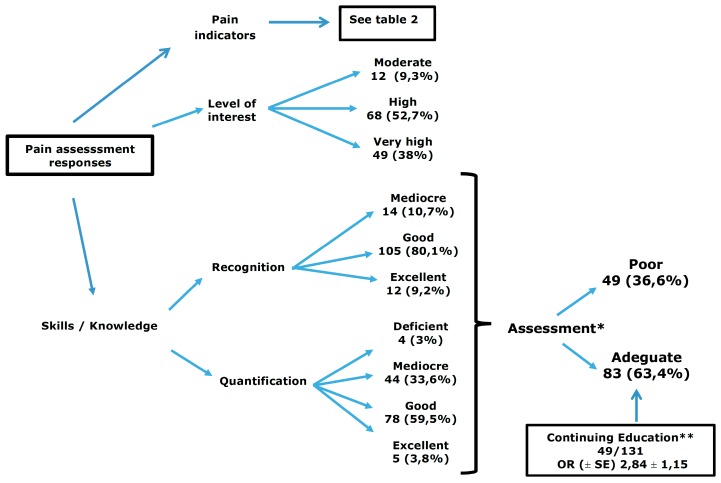
Pain assessment responses. * Pain assessment was arbitrarily deemed adequate when both “recognition” and “quantification” were excellent or good, and inadequate if one of these abilities was deficient, poor or mediocre. ** See section on results for further associations between pain assessment and different variables.

**Figure 2 vetsci-06-00006-f002:**
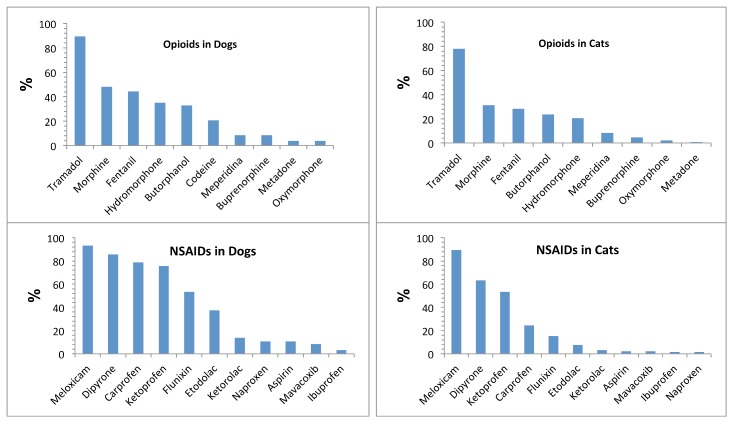
Frequency of non-steroidal anti-inflammatory drugs (NSAIDs) and opioids used in dogs and cats by Colombian veterinarians.

**Table 1 vetsci-06-00006-t001:** Demographic characteristics of 131 veterinarians interviewed in the city of Medellin to a survey of pain recognition and control in dogs and cats.

Characteristic	Distribution
Gender	76 males (58%)
	55 females (42%)
Age	
Between 23–35	73 (56.1%)
Between 36–50	38 (29.2%)
>50 years	19 (14.6%)
Type of practice	
Consulting office	43 (32.8%)
Clinic	60 (45.8%)
Hospital	28 (21.4%)
Years in practice	
2–10	83 (64.3%)
11–20	27 (20.9%)
>20	19 (14.7%)
Institution of graduation	
Univ. of Antioquia	101 (77%)
Others	32 (23%)

**Table 2 vetsci-06-00006-t002:** Ranking of pain indicators considered most useful in dogs and cats by 131 veterinarians in the city of Medellin, Colombia.

Pain Indicator	Number of Veterinarians Citing (%) *	
	Dogs	Cats
Abnormal postures or body position (tucked abdomen, arching, crouching postures, squinted eyes, sitting quietly seeking no attention)	89 (68%)	64 (50%)
Mobility (reluctance to move, lameness, slow pace)	70 (53.4%)	57 (45%)
Response to manipulation/palpation	69 (52.7%)	48 (38%)
Behavior/Activity (Unusual: “apathy, restlessness, aggression, anxiety, reduced interaction, hiding in cats”. Absence of normal: “lack of grooming in cats, reluctance to jump up, soiling outside the litter box”)	59 (45%)	82 (64%)
Anamnesis (history of signs provided by owner)	46 (35%)	46 (35%)
Physiological variables (heart rate, blood pressure, respiratory rate, temperature)	37 (28%)	33 (26%)
Vocalization (hauling, crying)	10 (8%)	17 (13%)
Appetite (inappetence)	10 (8%)	30 (24%)
Facial expression	2 (2%)	8 (7%)

* Each veterinarian was asked to choose three options from the list of nine possible pain indicators.

**Table 3 vetsci-06-00006-t003:** Evaluation of adequate knowledge on side effects of opioids in 131 veterinarians in the city of Medellin, Colombia.

Side Effect	Frequency (%) *	
	Dogs	Cats
Respiratory depression	48 (36.6%)	44 (33.5%)
Hypotension	38 (20%)	32 (24.4%)
Vomiting	27 (20.6%)	26 (19.8%)
Bradycardia	26 (19.8%)	14 (10.7%)
Nausea and/or panting	16 (12.2%)	17 (13%)
Sedation	15 (11.4%)	12 (9.2%)
Excitation	11 (8.4%)	20 (15.2%)
Gut hypomotility	11 (8.4%)	10 (7.6%)
Urinary retention	5 (3.8%)	5 (3.8%)
Convulsions	5 (3.8%)	7 (5.3%)
Diarrhea	2 (1.5%)	2 (1.5%)
Dysphoria	2 (1.5%)	11 (8.4%)

* Veterinarians were asked to choose only 2 of the most common side effects of opioids among 12 possible descriptors to assess knowledge on opioids side effects.

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
