# Peer review of "Survey of Pain Knowledge and Analgesia in Dogs and Cats by Colombian Veterinarians"

_vetsci, 2019, doi:10.3390/vetsci6010006_

Round 1
Reviewer 1 Report
line 33 couple of decades change into two decades.
line 231 pertaining the change into pertaining to.
line 238 replace 'about' with approximately.
line 316 what is AINE ? Please spell out.
line 324 after dosages - remove and, Mathews.
line 332 humans change into human.
line 335 replace to prescribe it in cats with they prescribed it for cats.
line 338 would consider a contraindication changed into considered.
line 339 inhibitor remove. and add as a contraindication.
line 346 remove were also indicated that .
line 347 ignore administered concurrently.
line 350 exists remove s.
line 352 remove resolve and replace with revolve
line 357 remove to and replace with of.
Also attached is an article on analgesics. With respect to tramadol, please add the comment that I have highlighted under tramadol, that refers to the lack of effect after 8 days, therefore, is of no value beyond this point.
The term 'acute renal failure' is now referred to as acute kidney injury (AKI).

Author Response
Dear reviewer,
1) All the suggested editorial changes have now been incorporated (deleted in BLUE, replaced in RED):
line 33 couple of decades change into two decades.
line 231 pertaining the change into pertaining to.
line 238 replace 'about' with approximately.
line 316 what is AINE ? Please spell out. mispelled now: NSAID
line 324 after dosages - remove and, Mathews.
line 332 humans change into human.
line 335 replace to prescribe it in cats with they prescribed it for cats.
line 338 would consider a contraindication changed into considered.
line 339 inhibitor remove. and add as a contraindication.
line 346 remove were also indicated that .
line 347 ignore administered concurrently.
line 350 exists remove s.
line 352 remove resolve and replace with revolve
line 357 remove to and replace with of.
2) "acute kidney failure" replaced by "acute kidney injury",
3) the following sentence was added in the discussion: In addition, a decrease of 60 to 70% in tramadol plasma concentrations occur after just 8 days of repeated administrations in dogs, making its long-term efficacy at best questionable (KuKanich, 2013).
4) Reference by Butch KuKanich has been added.
The authors are most obliged and grateful for the time spent to improve the style and content of the manuscript.
Best wishes for the new year.
Reviewer 2 Report
Dear authors,
I found this paper very interesting. It is a shame though it is very specific. I wonder if you could extends to other area of Columbia and therefore compare and have a broader picture of the knowledge and use of analgesia in Columbia. This is a great first marker and I hope future paper will show the work the vet in Columbia will put towards analgesia in pets as well as farm animal and others.
I only have a few comments or questions
In regards to the questionnaire page 2/10, I did not see a reference to human ethic. If you a ref, please put it in.
In the results, page 4/10, 3rd line starting with "Given " and finishing with "interest". I believe that sentence is more a discussion sentence and should therefore be moved.
Table 3 page 6/10. Add to title something referring to the number of vets and their origin: i.e. in 131 veterinarians in the city of Medellin, Colombia
Discussion
General comment:
Something missing I think in the discussion is the following. In the current study we are surveying mostly vets from Mendellin. I am not familiar with the area, but are the vets all from small animal practices or are there rural vets too? Some info is needed in the Results and then should be discussed with my following comment: My second point is that although we assume that the survey result could be extrapolated to the whole country, I would think it might be ok for big centres, but again what about rural areas...it would good for the authors to discuss briefly that aspects and that possibly the results may be different and how different by comparison to other survey???
Page 9/10, the final conclusion starting with "Taken" and finishing with "management": this is generalised to Columbia. I think it would be good to remind the reader that this survey targeted vet of a specific area of Medellin and that we are extrapolating the results to Columbian vets....and maybe to only big city vets????
Thank you again for this paper
Author Response
Dear reviewer,
Thanks for the careful review of the manuscript. In relation to the suggestions made, we have attempted to address them as precisely as possible.
1) In regards to the questionnaire page 2/10, I did not see a reference to human ethic. If you a ref, please put it in. Participation in the study was voluntary and all veterinarians were contacted ahead of visits to set appointments for the interviews.
2) In the results, page 4/10, 3rd line starting with "Given " and finishing with "interest". I believe that sentence is more a discussion sentence and should therefore be moved. This sentence has been deleted from the results section and is now in the discussion.
3) Table 3 page 6/10. Add to title something referring to the number of vets and their origin: i.e. in 131 veterinarians in the city of Medellin, Colombia. The Table title is now making that reference.
4)Discussion: General Comment
The discussion now has an ending that elludes to the study only involving veterinarians from the city of Medellin. Finally, as veterinarians were all registered within the urban area of Medellin, it will be of interest to extend this study to other cities and rural areas of Colombia.